Ecosystem services show variable responses to future climate conditions in the Colombian páramos

http://orcid.org/0000-0003-0448-5706 Diazgranados Mauricio 1 M.Diazgranados@kew.org
http://orcid.org/0000-0002-8256-9174 Tovar Carolina 2
http://orcid.org/0000-0002-3187-075X Etherington Thomas R. 2 3
http://orcid.org/0000-0002-2476-1598 Rodríguez-Zorro Paula A. 4
Castellanos-Castro Carolina 5
Galvis Rueda Manuel 6
http://orcid.org/0000-0001-6526-3037 Flantua Suzette G.A. 1 7 8 S.G.A.Flantua@gmail.com
1 Natural Capital and Plant Health Department, Royal Botanic Gardens, Kew , Ardingly, West Sussex , United Kingdom
2 Biodiversity Informatics and Spatial Analysis, Royal Botanic Gardens, Kew , Richmond, Surrey , United Kingdom
3 Manaaki Whenua – Landcare Research , Lincoln , New Zealand
4 Institut des Sciences de l’Évolution Montpellier (ISEM), Université de Montpellier , Montpellier , France
5 Ciencias Básicas de la Biodiversidad, Instituto de Investigación de Recursos Biológicos Alexander von Humboldt , Bogotá , Colombia
6 Departamento de Biología, Grupo de Investigación en Estudios Micro y Macro Ambientales (MICRAM), Universidad Tecnológica y Pedagógica de Colombia , Tunja , Colombia
7 Institute for Biodiversity & Ecosystem Dynamics, University of Amsterdam , Amsterdam , Netherlands
8 Department of Biological Sciences, University of Bergen , Bergen , Norway
Schuster Richard
Electronic publication date: 2021 May 3
Publication date: 2021
Volume: 9
Electronic Location ID: e11370
Received 2020 Nov 1; Accepted 2021 Apr 7
Copyright: © 2021 Diazgranados et al.
Copyright year: 2021
Copyright holder: Diazgranados et al.
License: This is an open access article distributed under the terms of the Creative Commons Attribution License, which permits unrestricted use, distribution, reproduction and adaptation in any medium and for any purpose provided that it is properly attributed. For attribution, the original author(s), title, publication source (PeerJ) and either DOI or URL of the article must be cited.
License URL: https://creativecommons.org/licenses/by/4.0/

Keywords: Boyacá, Climate change, Colombia, Mahalanobis distance, Niche modelling, Páramo, Useful plants

Funding: UK Department for Business, Energy & Industrial Strategy (BEIS) Royal Botanic Gardens, Kew BEIS National Royalties’ System (Sistema General de Regalías, SGR) Administrative Department of Science, Technology and Innovation of Colombia (Colciencias) European Research Council This work was supported by the UK Department for Business, Energy & Industrial Strategy (BEIS) and the Royal Botanic Gardens, Kew. Funds for this project were received from BEIS as part of a binational collaboration between the UK and Colombia, where Colombian partners are being funded by regional governments through the national royalties’ system (Sistema General de Regalías, SGR) and by the Administrative Department of Science, Technology and Innovation of Colombia (Colciencias). SGAF was additionally supported by the European Research Council under the EU Horizon 2020 Research and Innovation Programme (grant 741413 HOPE) Humans on Planet Earth–Long-term impacts on biosphere dynamics. The funders had no role in study design, data collection and analysis, decision to publish, or preparation of the manuscript.

==============================
Background

The páramos, the high-elevation ecosystems of the northern Andes, are well-known for their high species richness and provide a variety of ecosystem services to local subsistence-based communities and regional urbanizations. Climate change is expected to negatively affect the provision of these services, but the level of this impact is still unclear. Here we assess future climate change impact on the ecosystem services provided by the critically important páramos of the department of Boyacá in Colombia, of which over 25% of its territory is páramo.

Methods

We first performed an extensive literature review to identify useful species of Boyacá, and selected 103 key plant species that, based on their uses, support the provision of ecosystem services in the páramos. We collated occurrence information for each key species and using a Mahalanobis distance approach we applied climate niche modelling for current and future conditions.

Results

We show an overall tendency of reduction in area for all ecosystem services under future climate conditions (mostly a loss of 10% but reaching up to a loss of 40%), but we observe also increases, and responses differ in intensity loss. Services such as Food for animals, Material and Medicinal, show a high range of changes that includes both positive and negative outcomes, while for Food for humans the responses are mostly substantially negative. Responses are less extreme than those projected for individual species but are often complex because a given ecosystem service is provided by several species. As the level of functional or ecological redundancy between species is not yet known, there is an urgency to expand our knowledge on páramos ecosystem services for more species. Our results are crucial for decision-makers, social and conservation organizations to support sustainable strategies to monitor and mitigate the potential consequences of climate change for human livelihoods in mountainous settings.

Introduction

The Andean páramos occupy the highest ridges and plateaus of the Northern Andes–often called the Andean sky islands (Flantua et al., 2019, 2020), and are dominated by grasslands, rosettes and bushes, with an exceptionally high local endemism. Located above the upper forest line in the tropical Andes (3,000–3,200 m above sea level; m a.s.l.) and below the snowline, they are recognized as the world’s most diverse high-elevation ecosystem (Luteyn, 1999; Sklenář, Hedberg & Cleef, 2014; Rangel-Ch, 2000). They provide fundamental ecosystem services to communities on a local, regional and national scale. For instance, the páramos are considered the headwaters of South America by collecting, storing, and supplying freshwater (Buytaert, Cuesta-Camacho & Tobón, 2011; Nieto, Cardona & Agudelo, 2015) for an estimated 40 million people (Josse et al., 2009). Large cities, such as Bogotá and Medellín (Colombia), Quito and Cuenca (Ecuador), Piura and Cajamarca (Peru), depend on the páramos for over 80% of their fresh water supply (Rodríguez-Morales et al., 2014). In addition, the páramos function as carbon stocks (Hribljan et al., 2016; Maldonado, 2017), and provide cultural needs (Farley & Bremer, 2017), among other ecosystem services. Hundreds of the more than 4,700 páramo plant species are frequently used by locals as source of medicines, food, spices and condiments, construction materials, firewood, etc. (Torre et al., 2008). However, studies that describe the provision of services at a species level are scarce, e.g., pollination (Berry & Calvo, 1989) and hydrological functioning of Espeletia spp. (Cárdenas et al., 2018), and very few studies have addressed the question of how key ecosystem services in the páramos could be impacted by climate change.

The páramos are being negatively affected at an alarming pace due to changes in climate, land use strategies and over-exploitation (Castaño-Uribe, 2002; Anderson et al., 2011; Young, Young & Josse, 2011; Tovar, Seijmonsbergen & Duivenvoorden, 2013; Pérez-Escobar et al., 2018). Increased temperatures and changes in rainfall patterns have already caused an upward shift of the lower limit of the páramos in some areas (Morueta-Holme et al., 2015), and future projections suggest significant impacts on some of the most iconic plant species (Mavárez et al., 2018; Peyre et al., 2020; Valencia et al., 2020), and a possible reduction of its extent by c. 50% in the coming three decades (Tovar et al., 2013). As a result, it is projected that at the species level, between 10% and 47% of Andean endemic species could go extinct by the year 2100 (Malcolm et al., 2006). In addition, fluctuations in precipitation patterns and the loss of glaciers (Vuille et al., 2008, 2018; IDEAM, 2012) are expected to significantly affect water availability for the páramos and for human consumption, agriculture and energy generation, among other ecosystem services (Castaño-Uribe, 2002; Cuesta-Camacho, Peralvoco & Ganzenmüller, 2008; Chevallier et al., 2011; Buytaert, Cuesta-Camacho & Tobón, 2011; Young, Young & Josse, 2011). The rich variety of ecosystem services related directly to uses by local communities, such as provision of agrobiodiversity and wild harvest medicinal plants, and spiritual values, is under threat of a disruptive change of significant magnitude with future climate scenarios (Anderson et al., 2011).

Because of the paramount importance of páramo ecosystem services for livelihoods, it is imperative to quantify the likely impact of climate change on the páramo plants that supply ecosystem services. In this study, we analyse climate change impact on ecosystem services provided by the páramos of the department of Boyacá (Colombia), by associating regional uses of key species to the ecosystem services they provide, and by modelling their current and future potential distributions. We selected 103 key species based on their uses and ecological roles associated with ecosystem services and geographical extent. We hypothesise that trade-off patterns will be observed where certain ecosystem services will increase and other decrease. More specifically, we postulate that provisioning and regulating services will have the sharpest changes, both positive and negative (Alternative Hypothesis—Ha 1), related to changes in hydrological conditions expected to occur under warming climate conditions (Anderson et al., 2011; Buytaert, Cuesta-Camacho & Tobón, 2011). Additionally, we hypothesise that ecosystem services related to the provision of raw materials will possibly see an increase as the result of environmental changes (Ha 2), based on observed upward migration of species from lower elevations (Feeley et al., 2011; Zimmer et al., 2018), which are usually woodier (i.e., shrubs and trees) and have more biomass (Sevink et al., 2014). Therefore, woody taxa will likely benefit and expand their ranges to higher elevations (Young et al., 2017), potentially at the cost of the population size of herbaceous taxa (Feeley & Silman, 2010).

This study will contribute to understanding which ecosystem services could be most affected and at what magnitude, to inform decision makers and conservation/social organizations about the potential impacts on livelihoods. In addition, the results are aimed to be used by government and conservation organizations to register risks for natural capital assets, that will allow further monitoring and follow-up on future scenarios, as well as climate change mitigation and ecosystem-based adaptation for the future wellbeing of Andean communities.

Materials & methods

Study area

Colombia is known for its rich Andean topography of three mountain ranges (‘cordilleras’) that split in the southern area of the country with the Ecuadorian border and of which the Eastern Cordillera continues farther north towards the Venezuelan border (Fig. 1). Páramos are present in all cordilleras at elevations beyond the upper forest line, but the largest extent is found in the Eastern Cordillera. Here, the department of Boyacá has a prominent part of its territory covered by páramos (Fig. 1, 25%; 5,384 km2), with six páramo complexes (i.e., groups of páramos with shared biogeographic history), of which those of the Sierra Nevada del Cocuy and the Tota, Bijagual, Mamapacha complex are the largest, c. 2,000 and 1,500 km2 respectively. The area of Boyacá’s páramos represent 24% of Colombia’s páramos and 10% of the world’s páramos. However, only a quarter of the Boyacá páramos are protected under the national legislation of national parks (PNN) of which three are present in the department (Fig. 1). Despite the creation in recent years of various regional nature reserves and the government programme of delimitation of Colombian páramos, agricultural expansion, man-made fires and mining are notable threats to this ecosystem in Boyacá (Pérez-Niño & Leguizamón-Arias, 2020).

Figure 1 The department of Boyacá (Colombia), the study area and the main páramos.

PNN: National Nature Parks (‘Parques Nacionales Naturales’) of Colombia.

The páramos of Boyacá are extremely biodiverse: they currently hold 2,354 known plant and lichen species, classified in 806 genera and 243 families (Table 1). This is nearly half of the 4,646 species of plants and lichens reported for their entire department. Circa 1,500 are flowering plant species, and the families Asteraceae, Poaceae and Orchidaceae count for more than one third of the diversity (305, 141 and 76 species, respectively). Remarkably, 45 families have only one or two species, but some of them form extensive populations (e.g., Drimys granadensis L.f., Polylepis quadrijuga Bitter, Weinmannia tomentosa L.f.). The most diverse genera in the páramos of Boyacá are Espeletia (Asteraceae, 40 spp.), Miconia (Melastomataceae, 36 spp.), Elaphoglossum (Dryopteridaceae, 35 spp.), Campylopus (Dicranaceae, 32 spp.) and Cladonia (Cladoniaceae, 29 spp.). Most of the páramo species are still native (2,243 spp., 95%), and 399 are endemic to Colombia. Interestingly, although herbs account for most of these páramos diversity (917 spp.), there are also 86 species of trees, 422 of shrubs and 69 of vines and lianas. Despite the high diversity of these páramos, each area of continuous páramo has a relatively low number of species, many of them endemic to those areas. Therefore, páramos do not share many species, and very few species are broadly distributed. The number of plant species found in each páramo varies depending on its geographic extent, environmental range and climatic conditions, and species numbers vary from a few dozen to a few hundred, with fewer taxa shared among them (Jiménez-Rivillas et al., 2018; Londoño, Cleef & Madriñán, 2014; Urgiles et al., 2018).

Table 1 Richness of plants and lichens reported for Boyacá and its páramos.

Information extracted from the Colombian Catalogue of Plants and Lichens (Bernal, Gradstein & Celis, 2019).

Plant groups	Bryo-
phyta	Conife-
ropsida	Cyca-
dopsida	Lichens	Magno-
liopsida	Marchantiophyta	Pterido-
phyta	Grand
Total	
	Boyacá (all the ecosystems)	
Families	46	2	1	37	183	30	34	333	
Genera	129	3	1	77	1,191	85	107	1,593	
Species	293	3	1	263	3,369	247	470	4,646	
	Páramos of Boyacá (>3,000 m a.s.l.)	
Families	41	2	0	29	117	30	24	243	
Genera	107	3	0	53	507	68	68	806	
Species	246	3	0	180	1,500	180	245	2,354	
Native species	246	2	N/A	180	1,390	180	245	2,243	

We defined our study area by using a 1,000 m a.s.l. isoline that encloses the páramos of Boyacá and part of the Eastern Cordillera towards the north and south of the Boyacá department borders (Fig. 1; Fig. S2). Our study area encompasses lower elevations as several páramo species cover elevational gradients lower than the páramo itself. By doing this, we were able to use species occurrences in those elevations, to produce more robust species’ models and predictions (e.g., a species typical from lower elevations with narrower populations in the páramo may respond positively to climate change in this ecosystem, while a species typical from the páramo with no individuals found at lower elevations may display a negative response).

Database of uses and ecosystem services

To identify ecosystem services provision by páramo plant species, we first created a database of páramo plants of Boyacá with their uses, which were identified by reviewing international peer-reviewed literature (using online databases such as Science Direct, Web of Science and JSTOR), and national literature from universities, research institutes and local authorities and NGOs (Fig. 2; Table S1). Data sources were searched using five keywords: páramos, ecosystem services, uses, traits and Boyacá. We aimed specifically at literature describing uses in Boyacá, followed by literature describing uses in Colombia, having at the end 26 data sources which include books, scientific articles, reports, databases, guides and conference abstracts. Uses were then categorised following Cook (1995) using a level 1 and 2 classification (Table S2). Next, we defined a set of 19 ecosystem services grouped in 4 main categories, namely: (i) regulating (agroforestry, biological control, erosion regulation, pollination, restoration, water regulation), (ii) provisioning (food for animals, gene sources, food for humans, material, medicinal), (iii) supporting (barriers/windbreakers/support, conservation, nutrient cycling, ornamental resources, soil formation) and (iv) cultural (leisure, magic or religious, social), and assigned the identified uses to the different services.

Figure 2 Schematic overview of the project workflow.

The analysis required careful integration of plant uses recorded in the literature, plant occurrence data, and current and future climate data to produce estimates of how ecosystem services may change as a result of climate change.

Here we make three assumptions:Useful plants, as ecosystem service providers, can be used to infer the status of those ecosystem services. Therefore, it is considered that modelling the impact of climate change on plant species can inform the impact on each of the ecosystem services they correspond to (e.g., by modelling the impact on edible plants, it will be possible to estimate the impact on the ecosystem’s provision of food for humans).

Changes on the spatial breadth of the species’ environmental niche can be used to infer changes on the conservation status of the species’ populations, and consequently also on the magnitude of the ecosystem services provided. This implies that species populations cannot adapt to the new environmental conditions.

The collected information on species and ecosystems sufficiently represents the variation of ecosystem services found in páramo plants.

However, we do expect that there will be a relatively high contribution of ecosystem services related to medical uses, due to the larger proportion of medicinal plants in the ethnobotanical studies for the region (e.g., Lagos-López, 2007; Beltrán-Cuartas et al., 2010; Castellanos Camacho, 2011; Galvis, 2014) and because it is the most frequently reported plant use worldwide (Diazgranados et al., 2020a).

Species selection, species occurrences and climate data

We aimed to select more than 100 species from our compiled database of useful plants of Boyacá following first three main criteria per species:At least one use described within the Boyacá department.

The main species habitat is the páramo or is frequently reported in the páramo.

Low probability of taxonomic misidentification (based on our taxonomic experience).

We obtained a preliminary list of c. 130 species after applying criteria 1–3 and then collected geo-located records for those species from online databases (https://www.gbif.org/, http://biendata.org/, and https://sibcolombia.net/), to identify those with at least 20 records (4th criterion). We used only those records within the Americas and records falling in the sea or outside their assigned countries were eliminated. Additionally, elevation was calculated based on the provided coordinates using the global 90 m STRM (Jarvis et al., 2008) and compared with the elevation from the label of the specimen. Large differences were checked in Google Earth and species with numerous erroneous localities flagged for manual elimination. This resulted in a final list of 103 species (Table S3) having between 18 to 2,229 reliable locations. The species nomenclature followed the taxonomic backbone of ColPlantA (http://colplanta.org/; Diazgranados et al., 2020b).

The 103 species selected for this study comprise ten habits or growth forms: upright shrub (35.92% of the species), upright herb (20.39%), basal rosette (13.59%), tree (7.77%), prostrate herb (5.83%), stem rosette (5.83%), tussock (3.88%), prostrate shrub (2.91%), cushion/mat forming (1.94%) and trailing herb (1.94%) (Table S3). All the species are found in the páramos, either in the sub- and grass-páramo, approx. below 4,100 m a.s.l. (98.1%) or in the super-páramo, typically above 4,100 m a.s.l. (94.1%) (elevation belts sensu León, Jiménez & Marín, 2015). Most species (75.7%) extend their geographic range down to the Andean forest, below 3,000 m a.s.l. (e.g., shrubs from the sub-páramo that can be found in the Andean forest in lower frequencies; see also Fig 7. in Flantua et al. 2014), but only seven species of them thrive in them (e.g., trees that can be found indistinctly in either the high-Andean forest or the sub-páramo). Six species have not been reported in the super-páramo, and 25 species cannot be found in the Andean forest. In some cases, typical páramo species (e.g., Espeletia boyacensis Cuatrec., E. congestiflora Cuatrec.) can be found at lower elevations because of possible paramisation effect (i.e., transformation of the Andean forest into páramo vegetation after disturbance), the presence of azonal páramos (e.g., in poorly drained basins or high-Andean forest) or perhaps remnants from past distributions (Flantua et al. 2019).

For the climate data (~1960–1990) we used the 30 arc-second resolution 19 bioclimatic variables from WorldClim version 1.4 (Hijmans et al., 2005). For future climate projections we used the same biovariables for each available global circulation model (GCM), which provided predictions for: relative concentration pathway (RCP) values of 2.6, 4.5, 6.0, and 8.5; and for both the 2050 and 2070 time periods. Climatic values for all 19 bioclimatic variables were extracted for each species observation, and excluded repeated observations of the same species within the same climate cell such that there was only one species observation per climate cell.

Current and future climate niche modelling

We began our analyses by recognising that many of the 19 bioclimatic variables from WorldClim are highly correlated and that some of the variables may be less relevant to our study area. Principal components analysis (PCA) is a dimensionality reduction technique for eliminating redundancy of correlated variables (Pearson, 1901; Hotelling, 1933a; Hotelling, 1933b). This method has a long-standing tradition in niche studies for describing environmental space with a minimal set of independent axes (James, 1971). As PCA can be applied to raster geographic information system (GIS) surfaces (Demšar et al., 2013), it can be used, for instance, to create an environmental niche space from WorldClim data (Soberon & Nakamura, 2009). Here, we fitted a PCA for all 19 bioclimatic variables using the 57,665 1-km2 raster cells that defined our study area. In doing so we first pre-processed the climate data to ensure it was suitable for PCA (Nguyen & Holmes, 2019). We log-transformed all rainfall variables so that all 19 bioclimatic variables had unimodal and not overly skewed distributions. Then all variables were centred and scaled by transformation to z-scores. From our interpretation of the PCA loadings, the first three components primarily captured variation in temperature, moisture, and their combined seasonality, which together explained 86% of the climatic variance. We decided to limit our description of environmental space to the first three principal components as the fourth to nineteenth principal components only explained small amounts of further variance. Additionally, the variance explained by these components did not have a clear pattern, which would suggest that the variation being explained may be primarily noise in the data. All climatic data was then transformed into an environmental space defined by the first three principal components (see animated Fig. S2).

On the assumption that the actual distribution of species that are valuable for ecosystem services could be modified by humans through propagation and assisted dispersal, rather than trying to understand where species actually occurred, which would be affected by land use practices, we were solely interested in understanding at the broad spatial scale of our study area where the climatic conditions meant that they could occur. This question is best addressed by modelling the fundamental niche of a species that is “an n-dimensional hypervolume,…which corresponds to a state of the environment which would permit the species…to exist indefinitely” (Hutchinson, 1957, p. 416). Our choice of fundamental niche modelling methodology was heavily constrained by our species data. We did not have absence data, which precluded presence-absence methods. Also, due to the variety of data sources combined into our data set, we did not feel confident in defining a reliable background region to represent the area that was sampled, which precluded presence-background methods. Of the presence-only methods, we eliminated climatic envelope methods such as those based on convex hulls (Walker & Cocks, 1991), as these are sensitive to outliers which we expected might be present in our data. We also excluded climatic envelope methods as they produce binary niche estimates, but we expected that “there will however be an optimal part of the niche with markedly suboptimal conditions near the boundaries” (Hutchinson, 1957, p. 417). Of the presence-only techniques producing a continuous niche estimate we also eliminated density-based methods, as these require large samples free of sampling bias (Blonder et al., 2014), which our data was not. Ultimately, we decided that given our modelling question and data, the Mahalanobis distance was the most appropriate choice (Mahalanobis, 1936; Clark, Dunn & Smith, 1993; Farber & Kadmon, 2003; Etherington, 2019). As the Mahalanobis distance is based on a multivariate normal distribution, our models of fundamental niche would be elliptical in shape (see also Soberon & Nakamura, 2009). Also, although results vary by species, Mahalanobis distance has compared well against other presence-only, presence-background, and presence-absence modelling approaches (Dettmers et al., 2002; Johnson & Gillingham, 2005; Tsoar et al., 2007).

To explore issues of model uncertainty, we generated 1,000 bootstrap resamples (Efron, 1979; Diaconis & Efron, 1983) for each species as bootstrapping has been shown to be a reliable method of model evaluation (Verbyla & Litvaitis, 1989; Etherington & Lieske, 2019). Each bootstrap resample was then used to create a separate Mahalanobis distance model of the fundamental niche, with Mahalanobis distances converted to chi-square probabilities of being within the fundamental niche (Etherington, 2019). This resulted in each species having 1,000 different possible fundamental niches that varied in location, size, and orientation within environmental space (Figs. 3A–3C). Each bootstrap fundamental niche was then used to map within our study area each species’ potential niche, which is “the intersection of the fundamental niche with the realized environmental space at a particular time” (Jackson & Overpeck, 2000, p. 197). When done for the current climate data this meant that each species would have 1,000 maps of potential niche representing the probability that areas have climate conditions that could support the species. The total area of potential niche for each bootstrap resample was then calculated as the sum of all cells’ niche probability value (Fig. 4A). As the potential niche of a species within a region will change as the environment changes (Jackson & Overpeck, 2000; see animated Figs. S3, S4), the potential niche area for each bootstrap fundamental niche was also calculated for each combination of the 2050 and 2070 time periods and for the RCP 2.6, 4.5, 6.0, and 8.5 scenarios. For each combination of time period and RCP scenario, future climate predictions were selected randomly from the 10 GCMs that had data for all combinations (Figs. 4B–4I). Although we acknowledge that some GCMs may be more appropriate for our study area, we have followed the Intergovernmental Panel on Climate Change by assuming that each GCM is equally valid (Maslin, 2014), and as such we selected from the 10 GCMs randomly for each future potential niche. This process ultimately resulted in 1,000 estimates of potential niche area for each species for current climate conditions, and for the eight factorial combinations of the two future time periods and the four RCPS.

Figure 3 Example of Mahalanobis distance based fundamental niche modelling for bootstrap resamples 1, 2, and 1,000 of Espeletia boyacensis Cuatrec.

The bootstrap resample is shown by dots for which the size is indicative of the number of replicates in the sample. The probability of being within the fundamental niche is calculated from Mahalanobis distances for each sample and shown as ellipsoids at the p >= 0.9 (smallest ellipsoid), p >= 0.5, and p >= 0.1 (largest ellipsoid) thresholds for bootstrap resamples (A) 1, (B) 2, and (C) 1,000. For an animated version of this figure please see Fig. S1.

Figure 4 Example of potential niches for bootstrap resamples 1, 2, and 1,000 of Espeletia boyacensis Cuatrec.

Each subplot maps the potential niche probability for each bootstrap resample for (A) the present day, (B) 2050 under relative concentration pathway (RCP) 2.6, (C) 2070 under RCP 2.6, (D) 2050 under RCP 4.5, (E) 2070 under RCP 4.5, (F) 2050 under RCP 6.0, (G) 2070 under RCP 6.0, (H) 2050 under RCP 8.5, and (I) 2070 under RCP 8.5.

The bootstrap predictions of potential niche area were then combined by ecosystem service and converted into percentages of the study area. Changes in ecosystem service potential niche area between the present and all future climate combinations were presented as in terms of changes in percentage points. Preliminary analyses demonstrated that the distributions of potential niche areas were extremely complex and highly variable (Fig. 4), and therefore could not be reliably and consistently described using summary statistics. Therefore, we chose to use beanplots (Kampstra, 2008) as a method for visualisation and interpretation changes in potential niche area.

The initial geoprocessing of all data was programmed in Python (Pérez, Granger & Hunter, 2011) using the NumPy (van der Walt, Colbert & Varoquaux, 2011), GDAL (Warmerdam, 2008), and Pandas (McKinney, 2013) packages. Statistical analysis and plotting were programmed in R (R Core Team, 2018) using the beanplot (Kampstra, 2008), raster (Hijmans, 2018), fields (Nychka et al., 2017), RColorBrewer (Neuwirth, 2014), rgl (Adler et al., 2018), and misc3d (Feng & Tierney, 2008) packages.

Results

Uses and ecosystem services in Boyacá

We associated the uses of the 103 targeted species with 19 different ecosystem services at level 2 (ES2) for the four main level 1 ecosystem services (ES1) (Table S3; Fig. S5). ES1 with the highest and lowest number of ES2 is Regulation (n = 6) and Cultural (n = 3), respectively. The most common ecosystem services at level 2 (ES2) are Medicinal (n = 86) followed by Material (n = 37), Restoration (n = 28) and Ornamental resources (n = 26); the first two found in ES1 Provision, the latter two found in the ES1 Regulation and Support, respectively. A few ES2 were found only once or twice, such as Leisure in ES1 Culture, Genetic sources in ES1 Provision, and Agroforestry in ES1 Regulation. Species provide a median of two ES2 but some species provide up to eleven ES2 such as Macleania rupestris (Kunth) A.C.Sm. (shrub, creeper) and Vaccinium meridionale Sw. (shrub) from the Ericaceae family, and Morella parvifolia Benth. (shrub, tree) from the Myricaceae family (Fig. S5).

Bioclimatic change in Boyacá

We compared changes from present day to future conditions using all GCMs (Fig. 5). A comparison between the position of all the future ellipses and the directionality of the loadings for the bioclimatic variables shown as arrows indicates that although there is notable variation in individual GCM predictions, the trajectories of bioclimatic change are associated with increasing temperatures (BIO1, BIO5–11) and climate seasonality (BIO4 and BIO15). Therefore, as the RCP and time into the future increases, we would expect the study area to experience increases in minimum, mean, and maximum temperatures as well as greater temperature ranges, and temperature and rainfall seasonality including more extreme drier periods. These increasing temperatures and seasonality could both act as stressors on plant biology meaning that changes in potential niche will be expected.

Figure 5 Trajectories of bioclimatic change within the study area.

A principal component analysis (PCA) was applied to present-day bioclimatic variables to create three-dimensional climate space. The core areas (p = 0.75) of current and future climate are shown as ellipses. The current climate conditions are visualised by the black ellipse, with the 10 coloured ellipses visualising the core area of future climate conditions for each general circulation model (GCM) at 2070 under relative concentration pathway 8.5, which represents the most extreme future scenario. For an animated version of this figure please see Fig. S3.

Changes in potential niche area of plant taxa and ecosystem services

At a species level, potential niche area changes are highly variable but nearly all the 103 species appear more likely to decline than increase (see individual species beanplots and niche areas provided as Supplemental Files in figshare). Only two species look likely to experience an increase in potential niche area, Excremis coarctata (Ruiz & Pav.) Baker (herb) and Juncus effusus L. (herb). Other species such as Dryopteris wallichiana (Spreng.) Hyl. (fern), Myrsine coriacea (Sw.) R.Br. ex Roem. & Schult. (shrub, small tree), and Stevia lucida Lag. (shrub) could also experience a notable increase, but they could just as likely experience a decrease. Substantial losses even under the RCP 2.6 are likely for several succulent and bushy taxa, such as Echeveria quitensis (Kunth) Lindl., Eryngium humile Cav., Gaiadendron punctatum (Ruiz & Pav.) G.Don, and Vaccinium meridionale Sw.. The four Baccharis taxa show substantial changes. A few taxa show hardly any change, even under the most pessimistic scenarios, such as Espeletia incana Cuatrec. (stem rosette), Espeletiopsis pleiochasia (Cuatrec.) Cuatrec. (stem rosette), and Lycopodium clavatum L. (moss).

After aggregating species distributions into ecosystem service distributions, it becomes evident that the most likely outcome for most, if not all ecosystem services, is a decline in potential niche area. This decline increases as the severity of the RCP scenarios increases and as the time into the future increases and is up to around 20 percentage points, and possibly even around 30 percentage points, at both ES2 (Fig. 6) and ES1 (Fig. 7).

Figure 6 Beanplots of potential niche area change for level two ecosystem services under climate change.

Each bean represents the combined distribution of potential niche area change generated by 1,000 bootstrap resamples of a number (n) species associated with an ecosystem service. Beans are presented for the predicted climates at 2050 and 2070 under the four different RCP scenarios. Ecosystem services include: (A) cultural—leisure, n = 1; (B) cultural—magic or religious, n = 3; (C) cultural—social, n = 2; (D) provision—food for animals, n = 21; (E) provision—gene sources, n = 1; (F) provision—food for humans, n = 13; (G) provision—material, n = 37; (H) provision—medicinal, n = 84; (I) regulating—agroforestry, n = 1; (J) regulating—biological control, n = 4; (K) regulating—erosion regulation, n = 12; (L) regulating—pollination, n = 17; (M) regulating—restoration, n = 28; (N) regulating—water regulation, n = 15; (O) supporting—aesthetic values, n = 24; (P) supporting—barriers/windbreaks/support, n = 25; (Q) supporting—conservation, n = 3; (R) supporting—nutrient cycling, n = 3; (S) supporting—soil formation, n = 5.

Figure 7 Beanplots of potential niche area change for ecosystem services level one (ES1) under climate change.

Each bean represents the combined distribution of potential niche area change generated by 1,000 bootstrap resamples of a number (n) species associated with an ecosystem service. Beans are presented for the predicted climates at 2050 and 2070 under the four different RCP scenarios. Ecosystem services include: (A) cultural, n = 6; (B) provision, n = 157; (C) regulating, n = 77; (D) supporting, n = 60.

However, there were some notable differences in potential niche area changes under future conditions, especially at the ES2 level (Fig. 6). In cases such as Food for animals, Material and Medicinal, the range of changes is high and includes both positive and negative outcomes (Ha 1), while in other cases such as Gene sources, Agroforestry and Food for humans the large range is mostly substantially negative. In contrast, ecosystem services such as Leisure and Biological control have much smaller ranges of possible outcomes that are constrained to a reduction of around 0–10 percentage points, with extreme values reaching 30–40 percentage points. Also, the projections show a small possibility for a potential increase of a few percentage points for a few provision (e.g., Food for animals, Material (Ha 2), Medicinal), regulating (e.g., Erosion regulation, Pollination, Restoration, Water regulation), and supporting (e.g. Barriers/windbreaks/supports) services. Ecosystem services that are affected more by differences in the RCP scenarios are Gene sources, Agroforestry, Nutrient cycling, and Soil formation, which all show steep declines in potential niche area as RCP severity increases. Ecosystem services for which potential niche area declines more noticeable across time include Gene sources, Agroforestry, Erosion Regulation, Nutrient cycling, and Soil formation.

Discussion

The páramos of Boyacá are projected to experience higher temperatures and increased seasonality under climate change scenarios, and as a result, we show that significant changes are expected to happen by 2050, affecting the livelihoods of people depending on them. These changes will lead to an overall decrease in the provision of most ecosystem services (ES1) by native species (e.g., water regulation and materials). The magnitude of those changes is rather difficult to estimate, in part because of the lack of more precise information on the services provided (e.g., lack of data on income linked to goods/species), and in part because of the uncertainties regarding the plant responses to niche changes due to climate.

Although a few species are likely to increase their niche area, these seem not to have a great effect on the overall results at ES1 level. Tendencies of change are overly negative (mostly a loss of 10% but reaching up to a loss of 40%). However, some ecosystem services and taxa showed high variability in change (with both positive and negative possibilities), while those less variable were often negative. A few previous studies have highlighted the effects of climate change on ecosystem services provided by plants in the tropical Andes (Anderson et al., 2011), in the high Andean peatlands (e.g., Bounous et al., 2018) or in wetlands (e.g., Moor, Hylander & Norberg, 2015). A recent study concluded that the regions of Boyacá and Cundinamarca have experienced increased seasonality over the last 30 years (Fajardo Rojas, 2019), with direct negative impacts on farming yields (Tapasco et al., 2015). However, to our knowledge, this is the first study quantifying effects of climate change on ecosystem services for the páramos.

As we had expected, water regulation was among the ES2 with the highest projected losses (Ha 1). Among the plants providing this service is, for instance, the Puya goudotiana Mez (Bromeliaceae), whose rosette forms a water tank (Hornung-Leoni & Sosa, 2008) and is often suggested to contribute to the overall water storage of the páramos, as other tank-root species do in their habitats (Benzing & Bennett, 2000; Givnish et al., 2011; Males, 2016; Zotz et al., 2020). In general, inhabitants in close proximity to the páramos perceive páramo plants as providers of water provisioning service (Laverde Martínez, 2008; Farley & Bremer, 2017). Nevertheless, for many of the species related to this service, a clear relationship with water provisioning services remains to be analysed.

We also expected that the potential niche area of species providing materials would expand (Ha 2), given that warming climate is expected to favour thermophilic woody species (Lutz, Powell & Silman, 2013; Fadrique et al., 2018) which make up 89% of all materials in our dataset (Table S3). Yet, we mostly observed a decrease in potential niche area (Fig. 6), thus suggesting that rather than benefiting from increased temperatures these species could be suffering from other environmental changes. For instance, future climate projections suggest seasonality is likely to increase and extreme temperatures will be more frequent leading to a non-analogous climate to the current situation which imposes challenges to all taxa independent of life form. Only very few species such as the shrub Myrsine coriacea show an increase in potential niche area, but this tendency is insufficient to avoid the overall decrease projected for materials (Table 2). Among species with the highest projected losses are also those that provide the highest number of ES2. This is the case, for example, of the shrub Vaccinium meridionale, which currently provides 11 ES2, mostly provision (e.g., food for animals and people, materials, medicinal) and regulating services (e.g., erosion and water regulation, restoration) (e.g., Agrodiva Foundation, 2017).

Table 2 Examples of species recorded in the literature as providers of ecosystem services related to water regulation and materials.

Ecosystem service provided	Species	
Water regulation	Baccharis bogotensis Kunth, Calamagrostis effusa (Kunth)Steud., Carex pichinchensis Kunth, Clethra fimbriata Kunth, Escallonia myrtilloides L.f., Escallonia paniculata (Ruiz & Pav.) Schult., Hesperomeles goudotiana (Decne.) Killip, Hypericum juniperinum Kunth, Morella parvifolia (Benth.) Parra-Os., Nertera granadensis (Mutis ex L.f.) Druce, Paepalanthus dendroides (Kunth) Kunth, Puya goudotiana Mez, Vaccinium meridionale Sw., Vallea stipularis L.f., Weinmannia tomentosa L.f.	
Materials	Baccharis bogotensis Kunth, Baccharis latifolia Pers., Baccharis macrantha Kunth, Baccharis tricuneata Pers., Bejaria resinosa Mutis ex L.f., Bucquetia glutinosa DC, Calamagrostis effuse (Kunth)Steud., Cavendishia bracteata (Ruiz & Pav. ex J.St.-Hil.) Hoerold, Cestrum buxifolium Kunth, Chusquea tessellate Munro, Clethra fimbriata Kunth, Diplostephium rosmarinifolium Benth., Dodonaea viscosa Jacq., Drimys granadensis L.f., Duranta mutisii L.f., Escallonia paniculata (Ruiz & Pav.) Schult., Espeletia incana Cuatrec., Excremis coarctata (Ruiz & Pav.) Baker, Gaiadendron punctatum (Ruiz & Pav.) G.Don, Galium hypocarpium (L.) Endl. ex Griseb., Hesperomeles goudotiana (Decne.) Killip, Hypericum juniperinum Kunth, Jamesonia bogotensis H.Karst., Juncus effusus L., Macleania rupestris (Kunth) A.C.Sm., Miconia squamulosa Triana, Monnina salicifolia Ruiz & Pav., Morella parvifolia (Benth.) Parra-Os., Mutisia orbignyana Wedd., Myrsine coriacea (Sw.) R.Br. ex Roem. & Schult., Orthrosanthus chimboracensis (Kunth) Baker, Polylepis quadrijuga Bitter, Vaccinium floribundum Kunth, Vaccinium meridionale Sw., Vallea stipularis L.f., Viburnum tinoides L.f., Weinmannia tomentosa L.f.	

The ecosystem services provided under current climate conditions will inevitably change as a result of shifting base-line climate; estimating the scale and intensity of change, however, is complex and is defined at different levels of the páramo system. The first driver of change, and at the species level, is the loss of environmental space for a species providing an ecosystem service. As a consequence, the spatial distribution of a species is reduced, resulting in a net loss of individuals providing that particular ecosystem service.

The second driver of change is a change in the biotic interactions that operates at the level of the whole food web down to each unique biotic interaction. Shifts in, and perhaps decoupling of, biotic interactions are expected as a result of changes in the community disassembly, more so in tropical mountains (Sheldon, Yang & Tewksbury, 2011). For example, recent reports have documented new fungi-insect-plant interactions severely affecting local populations of páramo plants such as numerous species of Espeletia (Medina, Varela & Martínez, 2010). Also, it has been suggested that the introduction of invasive species associated with pastures (e.g., Anthoxanthum odoratum L., Poa annua L., Holcus lanatus L., and Cenchrus clandestinus (Hochst. ex Chiov.) Morrone) and agricultural systems (e.g., Rumex acetosella L. and Hypochaeris radicata L.) reduce the local biodiversity and the offer of ecosystem services (Gutiérrez-Bonilla et al., 2017).

The third driver of change is the result of the degree of functional compensation, or ecological redundancy, among species (see Rosenfeld, 2002a; Rosenfeld, 2002b). It might be that substantial loss in future ecosystem services provision might be less than expected due to several species playing equivalent roles in the ecosystems, i.e., they are ecologically redundant (Morelli & Tryjanowski, 2016). Our results show that the projected changes of ecosystem services in Boyacá are less extreme than previous studies suggest based on the expected reduction in general for Andean species (e.g., Ramirez-Villegas et al., 2014) and the páramo biome (e.g., Tovar et al., 2013). Given that an ecosystem service at ES2 is provided by several species, one explanation to the lower than expected change could be a certain degree of “compensation” or replacement between different species responses. Considering the extremely rich high elevation flora of the páramos, this could be a possibility. However, it is unlikely that species fulfil exact functional replicates of each other, and our understanding is far from grasping the complexity of biotic interactions on a multidimensional level of natural ecosystems (Morelli & Tryjanowski, 2016). Thus, the overall tendency indicates losses of ecosystem services across the different levels and scales, but these are less dramatic than previously suggested in large scale studies at species and biomes levels, but the exact causes are challenging to decipher.

Our results show that species are projected to suffer a reduction in their niche, which will likely affect floristic composition in the páramos of Boyacá and thus their ecosystem services. This is not surprising as evidence shows that upslope displacement of “alpine” species has occurred in the northern Andes in the last 200 years (Morueta-Holme et al., 2015; Moret et al., 2019). Also, Andean forest communities are already showing changes towards species with higher thermal optimum (thermophilisation) across the Andes in the last 15 years (Fadrique et al., 2018). Therefore, it is possible that thermophilisation might also be occurring in the páramo, however this remains to be tested. In the upper end of the páramo, colonization of newly deglaciated areas is occurring (Cuesta et al., 2019) but might show a lag in response (Zimmer et al., 2014; Zimmer et al., 2018). At the lower end, some species from the upper Andean forest could potentially be colonising the páramo and could provide some of the ecosystem services that were provided by the native species. However, some ES2 could be greatly affected by these compositional changes (Sheldon, Yang & Tewksbury, 2011) because specialization is thought to be greater in the tropics (Dyer et al., 2007). Asynchronous range shifts resulting in a community disassembly could thus, cause the introduction of novel interactions that either benefit or jeopardize the páramo’s functioning as a system (Larsen et al., 2011; Sheldon, Yang & Tewksbury, 2011). However, further analyses are needed on páramo species interactions in relationship to ecosystem services provided.

Our estimates of potential niche area change often had large ranges from around +10 to −40 percentage points at both ecosystem service level two (Fig. 6) and level one (Fig. 7). These large ranges of uncertainty could have resulted in part from variations in fundamental niche estimates resulting from bootstrap resampling (Fig. 3) that created variations in potential niche (Fig. 4A). A more significant source of uncertainty is likely derived from differences in future climate estimates from different GCMs which often produced very different future estimates of potential niches (Figs. 4B–4I). This is not surprising as mountain climates are hard to predict, and given the lack of guidance on choosing GCMs, it remains very important to use many GCMs to try and capture the full uncertainty of future climatic conditions (Beaumont, Hughes & Pitman, 2008). Also, the potential niche estimates could overestimate area as the realised niche resulting from biotic interactions such as competition will result in a realised niche that is smaller than the potential niche (Jackson & Overpeck, 2000). Likewise, it is important to note that the area of actual distribution will be much smaller again due to dispersal constraints limiting the areas that are or have been accessible to a species (Soberon & Nakamura, 2009). Also, while the scale of our modelling is appropriate given the extent of our study area, we acknowledge there will be microclimatic effects from topography operating below the 1 km2 grain of this study that will further affect the distribution of the potential niche at the scale of an individual plant (Lembrechts, Nijs & Lenoir, 2019).

Our literature review provided insights into the numerous ecosystem services provided by páramo species and those species that have a multi-functionality in terms of ecosystem services provision. The frequency of uses of the 103 species selected is somehow expected, i.e., most uses are linked to medicinal (n = 86), materials (n = 37), restoration (n = 28) and ornamental resources (n = 26). A similar pattern has been observed in the World (UPW) and Colombian (UPC) datasets of useful plants (Diazgranados et al., 2020a, 2020b). In our study, 83.5% of the selected species had medicinal uses, compared to 78.7% of the UPC and 66.2% of the UPW. Similar percentages are obtained for materials (35.6% this study, 32% UPC; 33.4% UPW); and environmental use (restoration plus ornamental resources; 27.2% this study, 27.5% UPC; 22.3% UPW).

However, some remarks are needed that further research could address building on the outcomes here presented. First of all, we collected and curated information for 103 species with corresponding georeferenced information. This number covers 23% of the estimated number of useful species hosted by the páramos of Boyacá (i.e., compared to 451 useful species reported for Boyacá’s páramos by the Instituto de Investigación de Recursos Biológicos Alexander von Humbodlt, 2014). Notwithstanding that it would be preferable to include more species, many individual páramo areas have between a few dozen and a few hundred species, with fewer being shared among them (Jiménez-Rivillas et al., 2018; Londoño, Cleef & Madriñán, 2014; Urgiles et al., 2018). Including more narrowly distributed species adds little weight to the provision of ecosystem services, and poses at least three challenges for modelling: (1) the scale of the climatic data precludes the inclusion of species with narrow distributions, as often they lack enough data points to inform the models; in fact, we had to discard 27 species because of redundant data points falling within the same grid cells of analysis; (2) narrowly distributed species often have very few georeferenced records; and (3) they cannot be used for finding patterns at larger scales, e.g., for the páramos of Boyacá.

In many cases, uses and their associated ecosystem services are unknown for species, thus relying on a still limited but slowly growing field of research. This inevitably results in a limited number of taxa per ecosystem service and a skew towards those ecosystem services most commonly studied, such as medicinal uses and materials (see also Diazgranados et al., 2020b). The addition of more species to the results here presented may as likely provide a more positive scenario of future predictions (by including more resilient species) as a more pessimistic scenario (by including more sensitive species). The results need to be interpreted in the context of the species included, and it is important to acknowledge that the results for ecosystem services with fewer species may be open to more change if many other species were added. However, only the inclusion of more species relying on a larger dataset and expanding literature may predict which way things could change.

For the páramos in particular, there is a daunting lack of understanding on the functionality of species within the community, let alone the functioning of the food web as a whole, precluding a better understanding of the species traits contributing to ES. Important contributions are currently being made to estimate páramo plant properties, such as freezing resistance, water relations, gas exchange characteristics (Rada, Azócar & García-Núñez, 2019), soil temperature regulations (Mora, Llambí & Ramírez, 2019), leaf water potential (Sandoval, Rada & Sarmiento, 2019), pollination (Pelayo et al., 2019), evapotranspiration (Rodríguez-Morales et al., 2019), dispersal strategies (Tovar et al., 2020), and biomass (Cabrera, Samboni-Guerrero & Duivenvoorden, 2018; Cabrera & Duivenvoorden, 2020). Though relevant base-line knowledge, such approaches alone will be a huge endeavour to cover a representative proportion of the c. 4,700 plant species currently identified for the páramo (Luteyn, 1999; Sklenář et al., 2005; Rangel-Ch, 2000). The multi-functional landscape of the páramos requires monitoring schemes of high technology (e.g. hyperspectral drone remote sensing; Borrelli et al., 2015; Zhang et al., 2020) and the inclusion of local knowledge (UICN, 2015), which need to be integrated at an overarching centralized level (see for instance the PARAGUAS project (2019–2021) (https://paraguas.ceh.ac.uk/)). Multi-disciplinary projects supported by strong collaborations between national universities and institutes should prioritize knowledge sharing through open-access platforms. An example is the digitized online system of biodiversity in Colombia (SIB, for Sistema de información sobre biodiversidad de Colombia; https://sibcolombia.net/), where also an increasing number of protocols are published, for instance to measure species traits.

Recommendations

To improve the livelihoods of people that depend on the páramo’s ecosystem services and thus are the first to suffer the consequences of climate change, several knowledge gaps need to be addressed. First of all, we are certain that the current lists of species known to occur in the páramos of Boyacá and Colombia are still incomplete, though major efforts have been undertaken recently (e.g., Diazgranados, 2012; Sylvester et al., 2019). Secondly, research focused on describing the functionality of species within the páramo community as a whole is pivotal but scarce. These gaps are at the frontline of scientific research but, considering the urgency and the expected velocity of temperature increase, outcomes from such research could potentially take decades to deliver and thus miss being included in the crucial development of current conservation management plans. Therefore, we outline several action items for prompt implementation by local governments (i–iii) and researchers engaged in studying the páramos under different climate conditions (iv):First, monitoring, monitoring and monitoring. It is of absolute critical value to monitor currently known functional properties of the páramo, even though we are not certain yet which taxa contribute most. This includes monitoring the productivity of the páramos in terms of hydrology (e.g. Cárdenas & Tobón, 2017). Permanent monitoring stations need to be installed and of access to the general public to allow capturing smaller changes occurring before a larger change is imminent and to help a community-based effort to adjust management plans when such changes are observed. Monitoring changes in plant composition and abundances in permanent plots are of crucial relevance to detect early warning signs of loss of taxa, invasive species and thermophilisation of the páramo. An expansion of the current network of GLORIA (https://redgloria.condesan.org; Cuesta et al., 2017) would provide the implementation of a consistent methodology over a larger region, but several methods are of equal value (Llambí et al., 2020). Additionally, a replicate study to this current study but implemented for the Andean forest will help to provide predictions whether species possibly migrating upwards to the páramo could be contributing or jeopardizing the provision of ecosystem services.

The involvement of local stakeholders to estimate how often people access the ecosystem services and how much resources are used. Building an open-access baseline database by the local communities of plant uses in relation to ecosystem services, not only contributes to monitoring essential taxa but also contributes to the establishment and recognition of the cultural heritage of a region.

Disseminate the knowledge and engage society about the importance of the páramo ecosystem and its plants in people’s lives. The monitoring efforts are unlikely to improve people’s livelihoods unless their results are clearly communicated, and results are centralised and made available to the public.

Ideally, fundamental niche models should be trained using not only present-day information, but also with information from the past for various reasons (Collevatti et al., 2013; Ordonez & Williams, 2013; Fitzpatrick et al., 2018). First of all, humans have altered the realized niche space for a range of different organisms (Albuquerque et al., 2018) either constraining (e.g., habitat destructions) or amplifying (e.g., invasive species), hampering the reconstruction of the fundamental niche space. Studies in the high tropics are few but such as those by Brugger et al. (2019) show the high potential legacy effect of past humans in high elevation Andean systems. However, more research in tropical ecosystems is needed to understand the possible subtle changes humans left in present-day species composition (McMichael, 2020). Secondly, palaeoecological reconstructions of the páramos have shown that the current interglacial, warm conditions are forcing these systems to be in a highly truncated spatial distribution in comparison to their ‘normal’ configuration, where the páramos occupied extensive area throughout the northern Andes (Flantua & Hooghiemstra, 2018; Flantua et al., 2019). For this same reason, it is likely that any study predicting future distributions of high mountain taxa are faced with a limited set of information to base the fundamental niche on. This has been flagged as critical in the usage of niche modelling (Maguire et al., 2015; Nogués-Bravo, 2009; Varela, Lobo & Hortal, 2011) but it would require a strong palaeoecological and macroecological approach which is currently still missing for the Andes, and many mountains around the world.

Conclusions

Our study represents one of the first attempts to quantify the impact of climate change on various ecosystem services of the páramos in general, by estimating the potential effects on plants contributing to them. The páramos of Boyacá are projected to experience higher temperatures and increased seasonality under climate change scenarios. As a result, species are projected to suffer a reduction in their potential niche, which will likely affect floristic composition in the páramos of Boyacá and thus the provision of ecosystem services by native species. The projected changes of ecosystem services in Boyacá are less extreme than previous studies suggest based on the expected reduction in general for Andean species and the páramo biome. Nevertheless, these changes may affect local inhabitants relying on them.

Tendencies of change are overly negative (mostly a loss of 10% but reaching up to a loss of 40%) with some ecosystem services and taxa showing high variability in change (with both positive and negative possibilities) with others less variable though often still negative. As we had expected, water regulation was among the ES2 with the highest projected losses. Contrary to what we had hypothesised, we mostly observed a decrease in potential niche area of species related to provision of raw materials, thus suggesting that rather than benefiting from increased temperatures these species could be suffering from other environmental changes (e.g., increased climate seasonality).

Some predicted impacts on the ecosystem services may change social and economic dynamics of local communities, transforming the region in unexpected ways and affecting livelihoods of people. Therefore, we urge government, academia and conservation organizations to create a baseline of the social and economic links to ecosystem services, and to register risks for natural capital assets. This will allow further monitoring and follow-up on future scenarios, as well as climate change mitigation and ecosystem-based adaptation for the future wellbeing of Andean communities.

Supplemental Information

Supplemental Information 1 Example of Mahalanobis distance based fundamental niche modelling for bootstrap resamples 1, 2, and 1,000 of Espeletia boyacensis Cuatrec.

The bootstrap resample is shown by dots for which the size is indicative of the number of replicates in the sample. The probability of being within the fundamental niche is calculated from Mahalanobis distances for each sample and shown as ellipsoids at the p>=0.9 (smallest ellipsoid), p>=0.5, and p>=0.1 (largest ellipsoid) thresholds for bootstrap resamples (A) 1, (B) 2, and (C) 1,000.

Click here for additional data file.

Supplemental Information 2 Visualisation of the climatic conditions of the study area in environmental and geographical space.

(A) A principal comment analysis (PCA) of the current climate conditions colour coded by axis dominance. The arrows show the direction and strength of the PCA loadings for the 19 bioclimatic variables. (B) It shows the same data in geographical space.

Click here for additional data file.

Supplemental Information 3 Trajectories of bioclimatic change within the study area.

A principal component analysis (PCA) was applied to present-day bioclimatic variables to create three-dimensional climate space. The core areas (p=0.75) of current and future climate are shown as ellipses. The current climate conditions are visualised by the black ellipse, with the 10 coloured ellipses visualising the core area of future climate conditions for each general circulation model (GCM) at 2070 under relative concentration pathway 8.5, which represents the most extreme future scenario.

Click here for additional data file.

Supplemental Information 4 Environmental and geographical space of the study area on current (top) and future (bottom) climate conditions.

Future scenario is for the year 2070 with RCP 8.5. Green ovals correspond to the core environmental niche of Espeletia boyacensis Cuatrec.

Click here for additional data file.

Supplemental Information 5 Distribution of uses and ecosystem services of the páramo plants of Boyacá, Colombia.

(a) Number of species by ecosystem services (at levels 1 and 2); (b) number of ecosystem services level 2 (ES2) per species, for species that provide more than five ES2.

Click here for additional data file.

Supplemental Information 6 Literature review on useful plants from the páramos of Boyacá.

Sources were searched using five keywords: páramos, ecosystem services, uses, traits and Boyacá.

Click here for additional data file.

Supplemental Information 7 Classification of plant uses according to Cook (1995).

Only levels 1 (upper case) and 2 (lower case) are displayed.

Click here for additional data file.

Supplemental Information 8 Selected plant species.

Species names and families are based on the taxonomic backbone of ColPlantA (http://colplanta.org/). Elevations and habits follow the Colombian Catalogue of Plants and Lichens (Bernal, Gradstein & Celis, 2019).

Click here for additional data file.

This project was part of the “Kew-Colombia Bio Programme” (2016–2020), which supports the development of primary research on biodiversity and ecosystem services in parts of Colombia with substantial voids in knowledge, to support the green growth of the country. We are very grateful to the Foreign & Commonwealth Office (FCO) - Colombia and the British Embassy in Colombia supported this programme by enabling meetings with stakeholders in both countries. We thank the Instituto de Investigación de Recursos Biológicos Alexander von Humboldt (IAVH) for their support and collaboration in Colombia, as well as the Universidad Pedagógica Tecnológica de Colombia (UPTC). The Instituto de Investigación de Recursos Biológicos Alexander von Humboldt (IAVH) and the Universidad Pedagógica Tecnológica de Colombia (UPTC) facilitated spaces for meetings and their staff provided advice on the methods. We are very grateful to Herman Amaya from the Gobernación de Boyacá, to the Corporación Autónoma Regional de Boyacá (CorpoBoyacá), to Parques Nacionales Naturales de Colombia, to Julián Barbosa from the Corporación Tibaira and to Maria Eugenia Morales from the UPTC, for facilitating various aspects of this project; to our local guides in the páramos of Boyacá: José Castellanos, William Gómez, José Ramiro García, Nicomedes Hernández and Juan Ponguta; to our students assisting us in the field and with the database: Andrea Simbaqueba, David Estéban Granados, Carlos Andrés Albarracín, Laura Pinzón, and Andrés Felipe Bohórquez; to Carolina Gómez Posada, Andrés Cuervo and Hernando García from the IAVH; and to Tiziana Ulian from RBG Kew, for her constant support. We also thank Elizabeth Aguilera and Georg Karl Weber from the Agrodiva foundation for their information about the uses of páramo plants.

Additional Information and Declarations

Competing Interests

Author Contributions

Data Availability

The authors declare that they have no competing interests.

Mauricio Diazgranados conceived and designed the experiments, performed the experiments, analyzed the data, prepared figures and/or tables, authored or reviewed drafts of the paper, and approved the final draft.

Carolina Tovar conceived and designed the experiments, analyzed the data, prepared figures and/or tables, authored or reviewed drafts of the paper, and approved the final draft.

Thomas R. Etherington conceived and designed the experiments, performed the experiments, analyzed the data, prepared figures and/or tables, authored or reviewed drafts of the paper, and approved the final draft.

Paula A. Rodríguez-Zorro analyzed the data, authored or reviewed drafts of the paper, contributed to gather information on plant uses from the literature, and approved the final draft.

Carolina Castellanos-Castro performed the experiments, analyzed the data, authored or reviewed drafts of the paper, and approved the final draft.

Manuel Galvis Rueda performed the experiments, analyzed the data, authored or reviewed drafts of the paper, provided knowledge on local plant uses, and approved the final draft.

Suzette G.A. Flantua conceived and designed the experiments, performed the experiments, analyzed the data, prepared figures and/or tables, authored or reviewed drafts of the paper, and approved the final draft.

The following information was supplied regarding data availability:

Data and R code are available at figshare: Diazgranados, Mauricio; Tovar, Carolina; Etherington, Thomas; Rodríguez-Zorro, Paula; Castellanos-Castro, Carolina; Galvis Rueda, Manuel; et al. (2020): Ecosystem services show variable responses to future climate conditions in the Colombian páramos - Data and Code. figshare. Dataset. DOI 10.6084/m9.figshare.13160684.v3.

The raw version of 1.4 WorldClim climate surfaces are available at http://www.worldclim.org/version1 and Surfaces for current are available at http://www.worldclim.org/current Please refer to the README.txt file in the worldclim folder (see figshare) for more information on their correct placement.

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
