# Peer review of "Ecosystem services show variable responses to future climate conditions in the Colombian páramos"

_PeerJ, doi:10.7717/peerj.11370_

## Round 0.1 · original submission · Major Revisions

We have now received two in-depth reviews of your study and you will see that both of them are positive toward your efforts and manuscript. I believe most, if not all of the reviewers' comments can be addressed with some changes in wording or rearranging of parts of your manuscript.

Please pay particular attention to the points raised by reviewer 2 in the 'Validity of the findings' section. It will be important in your response to reviewer comments to address these concerns.

Reviewer 1 ·

Basic reporting

This is a very well written manuscript and covers an important topic in an innovative fashion. Citations are fine, as is the organization and presentation of the findings.

Title is a bit succinct, as there are many high mountains in the Andes that do not have paramo and in fact are quite different. For that matter, there is paramo in Costa Rica, so not necessarily an Andean phenomenon.

More could be said about the effects of the assumptions described on page 6 of the ms.

Page 14 of ms talks of many other GCMs giving different findings, and illustrated in Figure 4. This seems like a lot of work done and very important to climate scientists, so could use some more text and explanation here.

I think the conclusions section is redundant and could be shortened.

Experimental design

I think the research design is fine, although as always, choosing just 100 or so species from a very rich flora seems arbitrary. I know this is always a criticism of those working in the tropics, and so it may be necessary and the authors seem aware of the seeming arbitrariness.

Why is the study area described as from 1000 m in elevation and above? Seems like data is from the paramo flora, at much higher elevations. Also, in general topography seems overlooked as a factor, given that higher up should be smaller areas and indeed eventually the topography will end and some species will go extinct due to mountaintop extinction. I was surprised topography was not part of the experimental design or the discussion.

Validity of the findings

Generally the more leaf area, the more evapotranspiration, and hence more water loss from the soil. So, the ecosystem service proposed for the Puya seems unlikely, and just an impression by people who have not quantified water and soil relations.

I think habitat is an important ecosystem service, as it provide refuge and food to many animals. This aspect does not seem to be captured in the ecosystem serviced and/or their discussion.

Additional comments

Well written and well done analysis. I think the title misleads a bit however, making it seem like it covers all the Andes of South America.

Reviewer 2 ·

Basic reporting

The hypotheses are clearly stated in the Introduction but only in the Conclusions do we clearly see what happened. I would suggest putting the main results of these hypotheses in the first paragraph of the Discussion so the reader can clearly understand what was expected and what happened.

Minor comments
1. Introduction – I think that the Intro could be tightened up. For instance, paragraphs 1 and 2 could be consolidated into one.
2. Line 82 – add a comma. “As a result, it is projected that at the species level, between 10 and 47% “.
3. It could be useful to number the hypotheses (lines 97, 101) and relate to them in the Discussion more clearly. They are only clearly mentioned in the Conclusions.
4. Line 107 – I suggest using the word “could” instead of “are going to”. These models and projections are still hypotheses of what “could” happen in the future.
5. Line 189 – low and high páramo… based on whose definition?
6. Line 190 – add dot in “m.a.s.l.”
7. Line 225 – Should Fig S1 be cited here? It was a little confusing. I think it is best to cite it where the Mahalanobis distance model is being described (along Fig. 3, line 255)
8. Line 306-307 – rephrase to “minimum, mean and maximum temperatures as well as …”.
9. Line 309 – I would, again, suggest using “could” instead of “will” since there are many other factors (e.g., biotic interactions, land use change) that could influence these trajectories.
10. Lines 313-322 – It’d be good to report percentages. What % of species show a decrease/increase/no change?
11. Line 323 – this is interesting but it should not be in the Results.
12. Line 325 – “most, if not all ecosystem services, is a decline…” (add commas).
13. Figure 6 and 7 legend – please say what the colors (orange, blue, green, yellow) mean.
14. Line 418 – rewrite to “some species from the upper Andean forest could potentially be colonizing”.
15. Recommendations i) and iii) could be consolidated since they are both about monitoring. If the authors want them separate, they should go in sequence.

Experimental design

1. The authors reviewed literature in order to derive the ES provision by páramo plants. However, they do not show how they did the search (keywords for international/indexed literature) or how many papers/documents they revised. I suggest explaining this more clearly and adding a supplementary material with the references (corresponding to what is described in lines 147 to 151).
2. Related to the previous comment, how were species placed in Cook’s categories? Only based on what is reported in the literature or did the authors include their own knowledge?

Minor comments
1. Line 178 – “several databases”. I think this has to be explained in more detailed. If not in the main document, the process should be described in Supplementary material.
2. Line 220 – seasonality of temperature or precipitation, or both? From Figure S2, since the axis is described as drought, I would guess seasonality of precipitation? But you refer to seasonality as a whole… Please clarify this and make it consistent (also in line 345).
3. I think it would be good to explain why the authors chose E. boyancensis in Figures 3 and 4. Given that it is a species that can be found in a greater altitudinal range (based on line 194-195), why did they choose that species as an example?
4. Line 437 – literature review should have a table in the Suppl. Material and a method describing how it was done.
5. Lines 450-460 – I think it is positive that the authors acknowledge the lack of data since this is a major issue in studying and understanding the potential effects of climate change.

Validity of the findings

1. Although the authors state in their first assumption (lines 158-160) that based on plant uses, ecosystem services can be inferred, I have some (conceptual) trouble with extrapolating from a few plant species to ES such as water regulation, erosion, nutrient cycling (n=3) and soil formation (n=5). Especially since the species number is low and I do not understand how can that be representative of an ES at a large scale (you are predicting at the regional level). I don’t disagree that individual species can have impacts on these ES but, are these numbers enough to talk about ES at the scale proposed? Should the authors maybe focus on ES with higher “n” and related to food, medicine, materials…?
2. Related to the previous comment, I’m also worried about drawing conclusions about Gene sources and Agroforestry when n=1 (also valid for cultural ES). For instance, Gene sources and Agroforestry are mentioned in the Abstract and Results (lines 332, 339, 341) as examples of ES with substantial negative responses… but with n=1 is it really valid to draw this conclusion?
3. Is there a criterion for a minimum number of species to reliably represent an ES?
4. I think it is important to remember that climate is just being defined by temperature and precipitation (and their variables). Climate, is much more comprehensive than just these two variables, especially for páramo plants where topography, soils, aspect, winds, etc. also play a crucial role. Therefore, in the paper (maybe in the assumptions/ discussion/conclusions) it should be acknowledged that projecting future climate change scenarios with just temperature and precipitation is still limited and that páramo responses may be much more complex due to all these factors.

Minor comments
1. Lines 378-383 – I think that while climate is important, climate here is only defined by temperature and precipitation. It is important to acknowledge that many other climatic variables are important (e.g., wind, topography, soils, etc.; WorldClim 2 has data for solar radiation, wind, water vapor pressure). I also consider that in some place in the discussion it should be acknowledged that these analyses are based on the current land use (which could also change, among others, with climate change).
2. Conclusions – how much do people actually use these species? There’s some emphasis on people suffering losses from reductions in SE (which is not unreasonable), but, are there are any studies/numbers supporting this claim?

Additional comments

This is an important study proving valuable information on ecosystem services (ES) of the páramos. It has a great amount of work on species modelling, ES, and the possible impacts of climate change (RCPs) on those services. I have not yet seen as study like this for the páramos, and I think this study is of relevance in our understanding of this key ecosystem. I also liked the Recommendations sections.
However, although the authors acknowledge their assumptions, as well as several of their analyses’ shortcomings, I have some areas of concern. I believe that addressing them could improve readability as well as understanding of this manuscript. I would like to apologize in advance, if I misunderstood certain areas of the manuscript.

---

## Round 0.2 · accepted · Accept

Thank you very much for submitting the revision of your manuscript. We have gotten feedback from one of the previous reviewers now and I have read the rebuttal and revised manuscript myself. They and I agree that your manuscript should now be accepted for publication in PeerJ.

Reviewer 1 ·

Basic reporting

Well written. The rewrite and the answers to previous reviewer comments were excellently and completely done.

All findings are well demonstrated; figures and tables all necessary.

In fact, there are still very worthwhile findings put into supplementary information. I agree with those designations, but always regret having an article split up into the main article and supplementary materials produced at great effort, but not read by typical users of the article.

Experimental design

Fine as done. This part was fine in previous submission and is now even better explained and options taken defended.

Validity of the findings

All is fine in the rewrite in regards these criteria.

Additional comments

Fine as is.